# Is metformin use associated with low mortality in patients with type 2 diabetes mellitus hospitalized for COVID-19? a multivariable and propensity score-adjusted meta-analysis

**Zhiyuan Ma●, Mahesh Krishnamurthy●***

Department of Medicine, St Luke's University Health Network-Easton Campus, Easton, Pennsylvania, United States of America

* Mahesh.Krishnamurthy@sluhn.org

**Data Availability Statement:** All relevant data are within the paper and its Supporting Information files.

## Abstract

### Background

Coronavirus disease 2019 (COVID-19) is a new pandemic that the entire world is facing since December of 2019. Increasing evidence has shown that metformin is linked to favorable outcomes in patients with COVID-19. The aim of this study was to address whether outpatient or inpatient metformin therapy for type 2 diabetes mellitus is associated with low in-hospital mortality in patients hospitalized for COVID-19.

### Methods

We searched studies published in PubMed, Embase, Google Scholar and Cochrane Library up to November 1, 2022. Raw event data extracted from individual study were pooled using the Mantel-Haenszel approach. Odds ratio (OR) or hazard ratio (HR) adjusted for covariates that potentially confound the association using multivariable regression or propensity score matching was pooled by the inverse-variance method. Random effect models were applied for meta-analysis due to variance among studies.

### Results

Twenty-two retrospective observational studies were selected. The pooled unadjusted OR for outpatient metformin therapy and in-hospital mortality was 0.48 (95% CI, 0.37–0.62) and the pooled OR adjusted with multivariable regression or propensity score matching was 0.71 (95% CI, 0.50–0.99). The pooled unadjusted OR for inpatient metformin therapy and in-hospital mortality was 0.18 (95% CI, 0.10–0.31), whereas the pooled adjusted HR was 1.10 (95% CI, 0.38–3.15).

**Funding:** The author(s) received no specific funding for this work.

**Competing interests:** The authors have declared that no competing interests exist.

## Conclusions

Our results suggest that there is a significant association between the reduction of in-hospital mortality and outpatient metformin therapy for type 2 diabetes mellitus in patients hospitalized for COVID-19.

## Introduction

Coronavirus disease 2019 (COVID-19), caused by the novel severe acute respiratory syndrome coronavirus 2 (SARS-CoV-2), is a new pandemic, leading to high mortality [1–5]. One of risk factors linked to worse outcomes for COVID-19 is preexisting type 2 diabetes mellitus [6]. Patients with diabetes hospitalized with COVID-19 are two- to three-fold more likely to be admitted into intensive care units than that of non-diabetics and the mortality rate is at least doubled [7–10].

There seems two distinct but overlapping pathologic phases in response to SARS-CoV-2 infection: the initial viral response phase triggered by the virus itself and the subsequent inflammatory phase triggered by the host response [11]. Accumulating evidence has shown cytokine storm syndrome in patients with severe COVID-19. Hence, agents that alleviate hyper-inflammation in COVID-19 patients could be beneficial. Metformin has been shown to modulate the immune responses and restore immune homeostasis in immune cells via AMPK-dependent mechanisms [12]. Historically, metformin was used during the treatment of influenza outbreak, due to its host-directed anti-viral properties [13]. Metformin has been widely used as the first-line therapy for type 2 diabetes mellitus. In light of the pathogenesis of SARS-CoV-2, several possible beneficial effects of metformin therapy in COVID-19 patients with pre-existing type 2 diabetes mellitus have been speculated, such as anti-inflammatory effects [14], reduction in neutrophils [15], increasing the cellular pH to inhibit viral infection, interfering with the endocytic cycle, reversing established lung fibrosis [16], and unique protective effects on microvessels [17]. Conversely, metformin therapy has also been challenged for the potential risk for lactic acidosis, particularly in the cases of multi-organ failure and for promoting SARS-CoV-2 infection by possibly increasing in ACE2 availability in the respiratory tract [18].

Accumulating evidence from retrospective studies suggests that treating pre-existing type 2 diabetes mellitus patients with metformin in COVID-19 patients may not be harmful, but even endows a protective effect and offers a low mortality. Due to the inherent nature of retrospective studies, biases are likely to confound the association between metformin therapy and outcomes. In this study, to limit the confounding bias, we aimed to address whether outpatient (prior to admission) or inpatient metformin (during hospitalization) therapy for type 2 diabetes mellitus is associated with low in-hospital mortality in patients hospitalized for COVID-19 by multivariable regression and propensity score-adjusted meta-analysis.

## Materials and methods

### Strategy of literature search for meta-analysis

A systemic literature search of studies published in English was performed in PubMed, Embase, Google Scholar and Cochrane Library up to November 1, 2022 using "((SARS-CoV-2) OR (Covid-19)) AND ((Metformin) OR (Biguanides) OR (Fortamet) OR (Glumetza) OR (Glucophage) OR (Biomet)) AND ((Mortality) OR (Death))". All abstracts were screened and the 'related articles' function was employed to broaden the search on appropriate abstracts

including meta-analysis. Full texts were then retrieved and reviewed. The meta-analysis was in adherence to the principles outlined by the Preferred Reporting Items for Systematic Reviews and Meta-Analyses (PRISMA). The study was registered on the PROSPERO (CRD42021284113).

## Study selection and risk of bias assessment for meta-analysis

Two reviewers independently extracted the following data from each study: first author, year of publication, study population characteristics (age and gender), study size, study design, country, definition of mortality, number of metformin users, effect size including odds ratios (ORs) or hazard ratios (HRs), confounding adjustment, and timing of metformin use. Studies included in the analysis met: 1) comparing the effect of metformin use prior to admission or during hospitalization on COVID-19 patients with type 2 diabetes; 2) the raw data on events can be extracted from the manuscript or supplementary materials or effect sizes (OR, HR) were reported; and 3) outcomes include in-hospital mortality. Studies were discarded, if: 1) duplicated studies from the same Registry or dataset; 2) studies without a control group or without in-hospital mortality reported. 3) total patients with type 2 diabetes in the study < 50; 4) studies that were not published in English. Discrepancies were resolved by discussion. The risk of bias in the included studies was assessed using the Newcastle-Ottawa Scale (NOS) [19].

## Data syntheses and statistical analysis

To assess whether metformin use is associated with low in-hospital mortality by comparing and integrating the results of different studies, meta-analysis was carried out as described previously [20]. OR was used as the main summary statistics. The OR represents the odds of a death event occurring in the metformin group in comparison to the non-metformin group. The point of estimate of the OR is considered statistically significant at the $P < 0.05$ level if the 95% confidence interval does not include the value 1. Raw event data extracted from individual study were pooled using the Mantel-Haenszel approach and effect size (OR, HR) was pooled using the inverse-variance method. In our study, random effect models were employed due to variance among studies. Heterogeneity between studies was investigated by the standard chi-squared Q-test. Publication bias was assessed by Funnel plots and *Egger's* test. A funnel plot is a type of scatter plot that resembles an inverted funnel (the 95% confidence interval), in which the treatment effects estimated from individual studies on the horizontal axis (OR), against a measure of study size on the vertical funnel (SE [log OR]). In addition, sensitivity analyses were performed by excluding one study at a time or subgroup to assess the robustness of the results. To better limit confounding bias due to baseline characteristics, several studies conducted analyses with propensity score matching approaches, in which propensity score was derived as the probability of being treated with metformin based on the participant's observed covariates between metformin and non-metformin users. All analyses were conducted with R (version 4.1.2) and R package 'meta'.

## Results

### Study selection and characteristics

After studies were carefully reviewed based on the inclusion and exclusion criteria, 22 retrospective observational studies were selected for further meta-analysis of metformin use and in-hospital mortality. The flowchart for the study selection is shown in Fig 1. Of these 22 studies, 17 studies reported on the association of outpatient metformin use and in-hospital mortality [21–37], 6 studies investigated the link between inpatient metformin use and in-hospital mortality [37–42], and 2 study probed the association of both outpatient and inpatient metformin

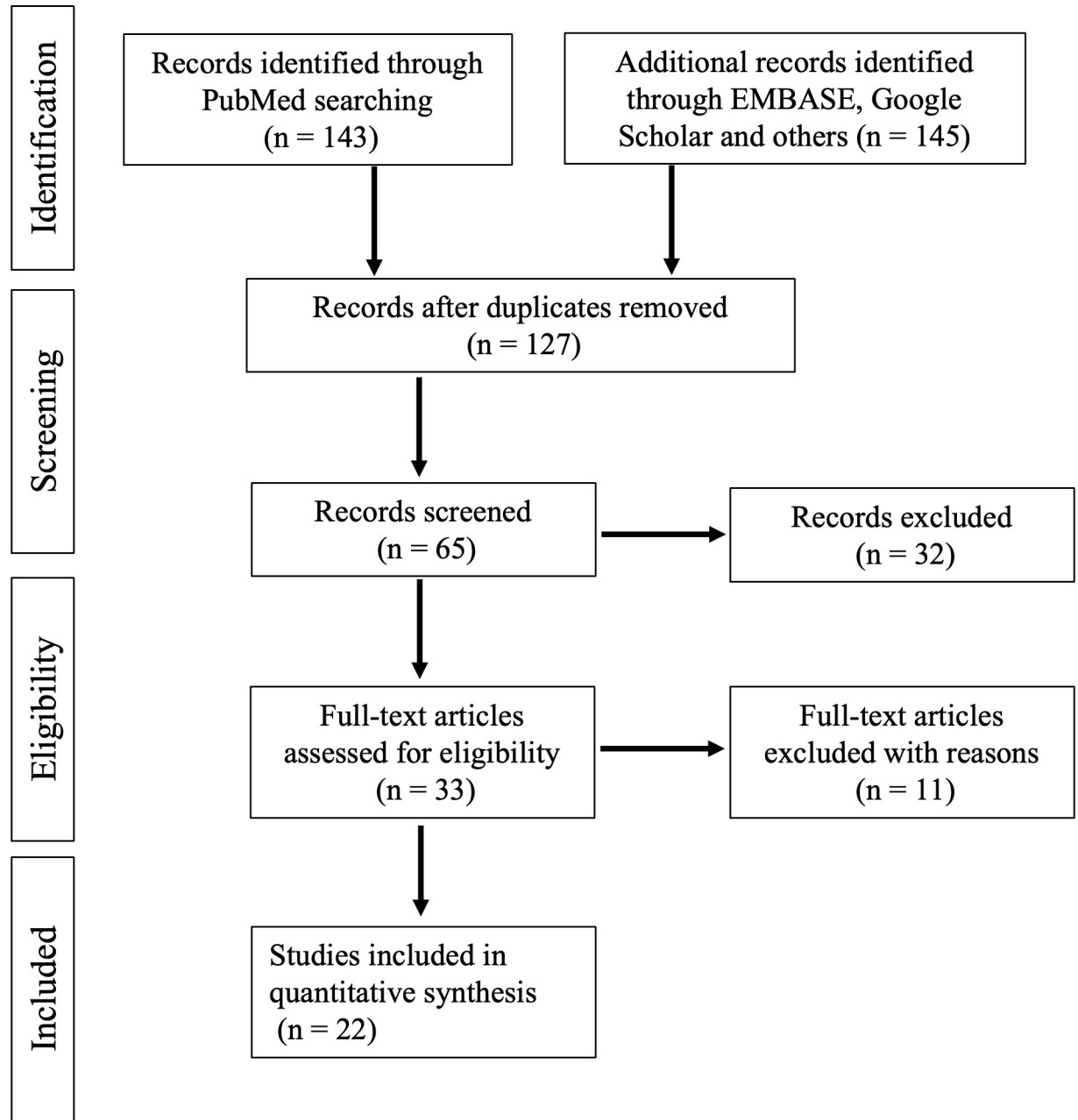

**Fig 1. Flowchart illustration of the study selection.**

use and in-hospital mortality [37,42]. There were 16 studies with extractable raw event data. According to adjustment for confounding bias, 16 studies performed multivariable regression or propensity scoring matching. The characteristics and quality assessment of the included studies are summarized in Table 1.

## Outpatient metformin use and in-hospital mortality

Due to the inherent nature of retrospective observational studies, to better probe the in-hospital mortality benefits and risks for outpatient metformin use, we set out to perform the meta-

**Table 1. Characteristics of studies comparing mortality in metformin users vs non-users in patients with type 2 diabetes hospitalized for COVID-19.**

| Study | Metformin (users/non-users) | | | Country | Endpoint | Confounding adjustment | Adjusted OR (95% CI)/ Note | NOS |
|---|---|---|---|---|---|---|---|---|
| | No. | Mean age (years) | Gender (% male) | | | | | |
| **Outpatient Metformin Use** | | | | | | | | |
| Studies with extractable raw event data | | | | | | | | |
| Abu-Jamous et at. 2020 [21] | 23/ 167 | NA | NA | UK | In-hospital mortality | Matching: age, sex, and number of admissions | 0.19 (0.05–0.70) | 7 |
| Bramante & Buse et al. 2021 [22] | 676/ 8879 | 60.4/ 54.2 (Median) | 56.2/ 46.6 | USA | mortality (in-hospital and before-hospital) | PS: age, race/ethnicity, gender, English-speaking status, T2DM, BMI category, history of bariatric surgery, NAFLD)/NASH, CAD, heart failure, CKD, hypertension, hyper- or hypo-coagulable state, interstitial lung disease, tobacco use, and home medications | 0.38 (0.16–0.91)/ MSA/ BMI > 25 kg/m$^2$ | 8 |
| Bramante & Ingraham et al. 2021 [23] | 2333/ 3923 | 73.0/ 76.0 | 51.6/ 44.6 | USA | In-hospital mortality | Cox proportional hazards and PS: age, sex, comorbidities, alcohol abuse, HIV, asthma, inflammatory bowel disease, dementia, Charlson comorbidity index, and medications, and state | Adjusted HR: 0.89 (0.78–1.01) OR: 0.91 (0.78–1.07)/MSA | 8 |
| Cariou et al. 2020 [24] | 746/ 571 | 70 (both groups) | 65 (both groups) | Franch | In-hospital mortality | | Duplicated report from the CORONADO study as Lalau et al. 2020/ not included for analysis | 7 |
| Chen et al. 2020 [25] | 43/ 77 | 62.0/ 67.0 | NA | China | In-hospital mortality | | | 6 |
| Crouse et al. 2020 [26] | 76/ 144 | NA | NA | USA | In-hospital mortality | | | 6 |
| Ghany et al. 2021 [27] | 243/ 350 | NA | NA | USA | In-hospital mortality | Cox proportional hazards: age, gender, Charlson score, diabetes, hypertension and ejection fraction | Adjusted HR: 0.74 (0.53–0.98) | 8 |
| Lalau et al. 2020 [28] | 1496/ 953 | 68.5/ 74.6 | 66.8/ 59.6 | France | In-hospital 28-day mortality | IPTW: sex, age, BMI, arterial hypertension, history of disease, active cancer, treated OSA, use of any of antidiabetic drugs | 0.71 (0.54–0.94)/ MSA | 8 |
| Li et al. 2020 [29] | 37/ 94 | 64.6/ 67.7 | 59.5/ 55.3 | China | In-hospital mortality | | | 6 |
| Luk et al. 2021 [35] | 737/ 254 | 65.6/ 68.9 | 55.0/ 51.6 | China | In-hospital mortality | Adjusted for age, sex, smoking, diabetes duration, HbA1c level, comorbidities (hypertension, CAD, heart failure, cerebrovascular disease, CKD, chronic liver disease, COPD), preadmission use of other glucose-lowering drugs, statins and RAAS inhibitors, and in-hospital use of other glucose-lowering drugs. | Adjusted HR: 0.51 (0.27–0.97) | 8 |
| Ma et al. 2022 [30] | 361/ 995 | NA | 60.4/ 54.1 | USA | In-hospital mortality and hospice | Multivariable, PS and IPTW: age, height, weight, gender, race, comorbidities (CHF, CAD, asthma, COPD), clinical lab tests (BUN, creatinine, ALT, total bilirubin) | 0.25 (0.06–0.74) PS: 0.22 (0.04–0.81) IPTW: 0.25 (0.07–0.89)/MSA | 8 |
| Mirani et al. 2020 [31] | 69/21 | 69/ 75 | 72.5/ 71.4 | Italy | In-hospital mortality | Adjusted for age and sex | Adjusted HR: 0.55 (0.27–1.11) | 7 |
| Ojeda-Fernandez et al. 2022 [36] | 23327/ 8639 | 70.7/ 75.0 | 56.7/ 60.9 | Italy | In-hospital mortality | PS: age, sex, duration of diabetes, DDCI and co-morbidities, and medications of interest | PS: 0.74 (0.67–0.81) | 8 |
| Ong et al. 2021 [37] | 109/ 169 | NA/ 63.9 | 62.4/ 52.7 | Philipp-ines | In-hospital mortality | Multivariable model: age, CKD, ACS, Tocilizumab, Systemic steroids, Convalescent plasma therapy, Hemoperfusion, In-hospital insulin use, HbA1c | 0.43 (0.23–0.82)/ including inpatient metformin use | 7 |

(*Continued*)

**Table 1.** (Continued)

| Study | Metformin (users/non-users) | | | Country | Endpoint | Confounding adjustment | Adjusted OR (95% CI)/ Note | NOS |
|---|---|---|---|---|---|---|---|---|
| | No. | Mean age (years) | Gender (% male) | | | | | |
| Pérez-Belmonte et al. 2020 [33] | 825/ 663 | 74.8/ 77.1 | 65.7/ 57.2 | Spain | In-hospital mortality | PS: age, gender, history of smoking, hypertension, dyslipidemia, CKD, cerebrovascular disease, COPD, atrial fibrillation, CAD, heart failure, obesity, dementia, Barthel Index score, and Charlson Comorbidity Index score, treatment with ACEi, ARB, anticoagulant, and statin, admission blood glucose, serum creatinine, and transaminase levels | 1.16 (0.78–1.72)/ MSA | 8 |
| Ramos-Rincón et al. 2020 [34] | 420/ 370 | NA | NA | Spain | In-hospital mortality | Multivariable analysis: demographics (age, sex, acquisition), comorbidities and dependence, symptoms (dyspnea), physical examination, laboratory findings and treatment | 0.98 (0.64–1.79)/ patients ≥80 years | 8 |
| Tamura et al. 2021 [42] | 116/ 72 | 62.1/ 68.6 | 62.1/ 63.9 | Brazil | In-hospital mortality | | | 7 |
| **Studies with only pre-calculated effect size (OR, RR or HR) data** | | | | | | | | |
| Oh et al. 2021 [32] | 2047 (total) | NA | NA | South Korea | In-hospital mortality | Multivariable model: age, sex, place of residence, disability, income, the Charlson Comorbidity Index, other anti-diabetic medications | 1.26 (0.81–1.95) | 8 |
| **Inpatient Metformin Use** | | | | | | | | |
| **Studies with extractable raw event data** | | | | | | | | |
| Jiang et al. 2021 [39] | 100/ 228 | 64.0/ 67.0 | 49.0/ 54.8 | China | 30-day all-cause mortality | PS: age, gender, weight, FBG, severity of COVID-19, Charlson comorbidity index, CHD, metformin therapy prior to hospitalization, DDI, creatinine and site | Adjusted HR: 0.54 (0.13–2.26)/ MSA | 8 |
| Li et al. 2022 [40] | 37/ 94 | NA | NA | China | In-hospital mortality | | | 6 |
| Luo et al. 2020 [41] | 104/ 179 | 63.0/ 65.0 | 51.0/ 57.5 | China | In-hospital mortality | | | 6 |
| Ong et al. 2021 [37] | 40/ 169 | NA/ 63.9 | 65.0/ 52.7 | Philipp-ines | In-hospital mortality | | | 7 |
| Tamura et al. 2021 [42] | 115/ 73 | 63/ 67 | 63.5/ 61.6 | Brazil | In-hospital mortality | Multivariable analysis: In-hospital metformin therapy, Pre-hospital metformin therapy, In-hospital insulin therapy, Dexamethasone/ prednisolone | 0.034 (0.002–0.58) | 7 |
| **Studies with only pre-calculated effect size (OR, RR or HR) data** | | | | | | | | |
| Cheng et al. 2020 [38] | 678/ 535 | 62.0/ 64.0 (Median) | 53.8/ 49.9 | China | In-hospital 28-day mortality | Cox proportional hazards and PS: age, gender, comorbidities, blood glucose, C-reactive protein, estimated glomerular filtration, alanine aminotransferase, and creatinine | Adjusted HR before PS: 0.87 (0.36–2.12) Adjusted HR after PS: 1.65 (0.71–3.86)/MSA | 8 |

PS: Propensity score; IPTW: Inverse probability of treatment weighting; NA: Not accessed; MSA: Metformin-specific adjustment (The probability of a patient being treated with metformin as dependent variable was estimated using a logistic regression model); ACS, acute coronary syndrome; CHF, congestive heart failure; CAD, coronary artery disease; CKD, chronic kidney disease; COPD, chronic obstructive pulmonary disease; DCCI, Drug-Derived Complexity Index; ALT, alanine aminotransferase; BUN, blood urea nitrogen; TBILI, total bilirubin; NAFLD: Nonalcoholic fatty liver disease; NASH: Nonalcoholic steatohepatitis; NOS: The Newcastle-Ottawa Scale.

analysis using two approaches. First, meta-analysis of 16 studies with extractable raw event data from 30,891 metformin and 25,770 non-metformin users revealed that outpatient metformin use was associated with a significant reduction of in-hospital death (Fig 2A) with a pooled unadjusted OR of 0.48 (95% CI, 0.37–0.62; $I^2$ = 87%). Publication bias was assessed using a

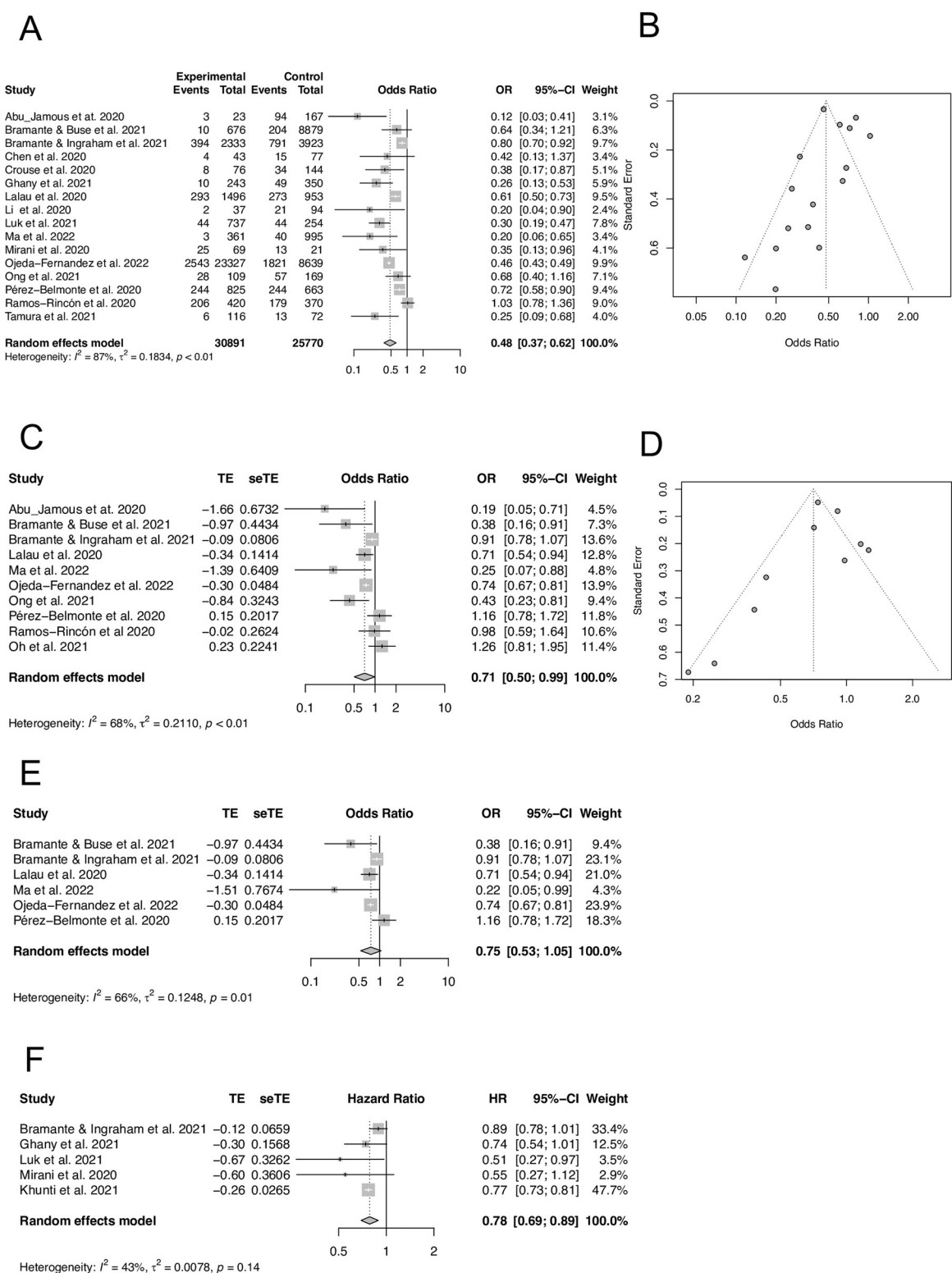

**Fig 2. The association of outpatient metformin use and in-hospital mortality in patients with COVID-19 and pre-existing T2DM.**
(A) Meta-analysis of 16 selected studies with extractable raw event data using the Mantel-Haenszel approach. (B) Funnel plot test of 16 studies included in panel A. (C) Meta-analysis of 10 selected studies with multivariable or propensity score-adjusted OR using the inverse-variance method to pool effect sizes. (D) Funnel plot of 10 studies included in panel C. (E) Meta-analysis of OR for in-hospital mortality in 6 studies with propensity score matched for metformin use by the inverse-variance method. (F) Meta-analysis of 5 studies with adjusted HR for in-hospital mortality using the inverse-variance method.

funnel plot (Fig 2B) which showed the 16 studies used in our meta-analysis. *Egger's* test was performed to quantitively assess publication bias and revealed no significant publication bias (p = 0.86). Second, meta-analysis of 10 studies with adjusted OR using either multivariable regression or propensity score matching demonstrated a pooled OR of 0.71 (95% CI, 0.50–0.99; $I^2$ = 68%), suggesting that there was a significant association between decreased in-hospital mortality and outpatient metformin use (Fig 2C). The funnel plot for the included 10 studies showed an approximate symmetric distribution (Fig 2D). *Egger's* test revealed no statistically significant publication bias (p = 0.54).

In the subgroup analysis of propensity scored-matched studies, outpatient metformin use was not associated with a statistically significant but a trend toward a reduction of in-hospital mortality with a pooled OR of 0.75 (95% CI, 0.53–1.05; $I^2$ = 66%) (Fig 2E). In contrast, subgroup analysis of 5 studies with adjusted HR revealed outpatient metformin use was associated with a statistically significant decrease of in-hospital mortality with a pooled HR of 0.78 (95% CI, 0.69–0.89; $I^2$ = 43%) (Fig 2F). In the subgroup analysis of studies with extractable raw events, excluding studies with NOS < 7 or sample size < 100 revealed a similar significant association between outpatient metformin use and in-hospital mortality (OR 0.50; 95% CI, 0.37–0.68; $I^2$ = 90%) (S1 Fig). In addition, to determine the impact of individual studies on pooled effects, sensitivity analysis was performed by excluding one study a time. The pooled ORs in studies with extractable raw event data showed constantly significant differences with the upper limit of 95% CI less than 1 (S1B Fig), whereas studies with adjusted OR demonstrated unstable results in the sensitivity analysis (S1C Fig). These results indicate that outpatient metformin use is associated with a significant reduction of in-hospital death in unadjusted raw data with stability of meta-analysis and that the association was still statistically significant after adjustments for confounding but with instability in the sensitivity analysis.

## Inpatient metformin use and in-hospital mortality

There were a few studies mainly from China investigating the association between inpatient metformin use and in-hospital mortality in patients with type 2 diabetes hospitalized for COVID-19. We performed the similar analyses with extractable raw data and adjusted OR. Meta-analysis of 5 studies with extractable raw event data from 396 metformin and 743 non-metformin users showed that inpatient metformin use was associated with a significant reduction of in-hospital death (Fig 3A) with a pooled unadjusted OR of 0.18 (95% CI, 0.10–0.31; $I^2$ = 0%). The funnel plot for the included 5 studies demonstrated a good symmetric distribution (Fig 3B). However, meta-analysis of 2 studies with adjusted HR demonstrated a pooled HR of 1.10 (95% CI, 0.38–3.15; $I^2$ = 43%), suggesting there is no statistically significant association between inpatient metformin use and in-hospital mortality (Fig 3C). No sensitivity or subgroup studies were performed due to small number of studies available.

## Discussion

Diabetes has been linked to poor outcomes in patients with COVID-19, so treatments that are effective, safe, and available immediately are urgently needed. Metformin is a the first-line therapeutic agent that is safe, effective, and inexpensive for type 2 diabetes mellitus. In the light of host-directed anti-viral properties and anti-inflammatory effects of metformin, many retrospective studies including our previous study [30] has explored whether metformin is beneficial in patients with COVID-19 and preexisting type 2 diabetes mellitus. To further compare and integrate the results of different studies and to increase statistical power, we set out to perform a meta-analysis. In this meta-analysis study, we found that outpatient metformin use was associated with a significantly lower in-hospital mortality in patients with type 2 diabetes

## A

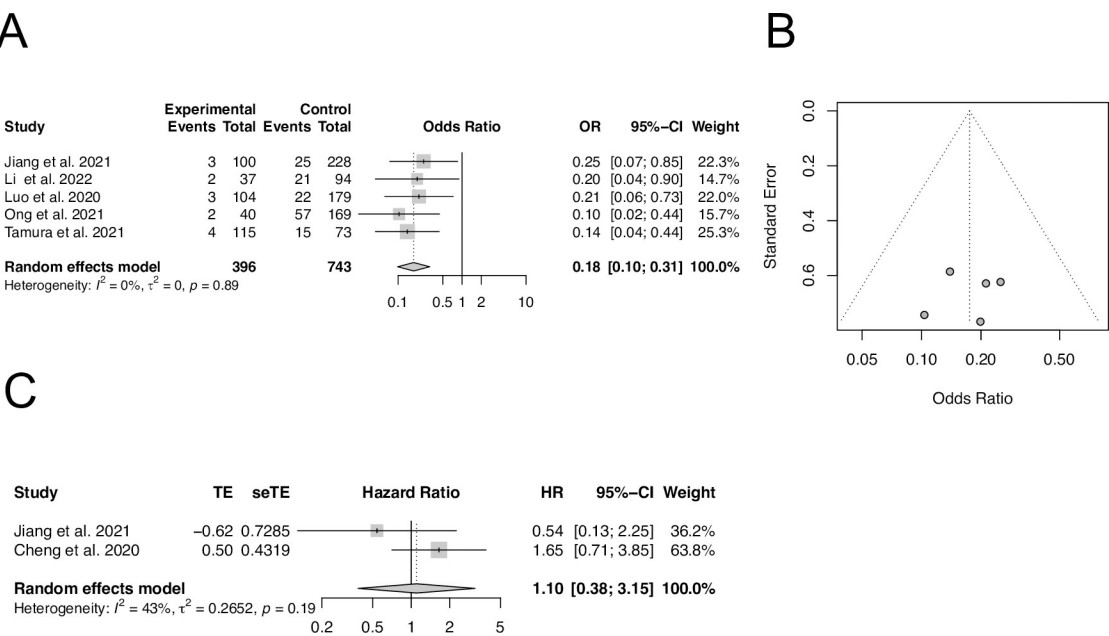

## B

## C

**Fig 3. The association of inpatient metformin use and in-hospital mortality in patients with COVID-19 and pre-existing T2DM.** (A) Meta-analysis of 5 selected studies with extractable raw event data using the Mantel-Haenszel approach. (B) Funnel plot of 5 studies included in panel A. (C) Meta-analysis of 2 selected studies with multivariable and propensity score-adjusted HR using the inverse-variance method to pool effect sizes.

hospitalized for COVID-19 in the raw and adjusted analyses, compared to non-metformin use. We also found that there was a significant association between decreased in-hospital mortality and inpatient metformin use in the unadjusted analysis, but this significant association was not observed after adjustments for confounding factors.

Prior to this meta-analysis, several meta-analyses [43–45] of the association between metformin therapy and severity and mortality were published, but they have issues. First, metformin therapy was not differentiated between outpatient and inpatient use, as oral hypoglycemic medications including metformin are usually held during hospitalization. Second, while study endpoints such as pre-defined mortality were somewhat different among studies, effect sizes (OR or HR) of different mortality outcomes were combined in previous meta-analysis studies. Third, due to nature of retrospective studies, various effect sizes with adjustments for baseline characteristics were reported in several studies. However, unadjusted and adjusted effect sizes were mixed in different meta-analyses. Fourth, using pre-calculated standard error instead of raw event data may not be a good estimator of the precision of the binary effect size (OR or HR). Nevertheless, only pre-calculated OR was used in the previous meta-analyses.

In this study, we specifically addressed these issues by investigating the association between outpatient or inpatient metformin therapy and in-hospital mortality in patients with diabetes hospitalized for COVID-19. We also used extractable raw data instead of pre-calculated ORs or HRs to determine the pooled OR for the unadjusted analysis. We found that there was a significant association between the reduction of in-hospital mortality and outpatient or inpatient metformin therapy for type 2 diabetes mellitus in COVID-19 patients in the unadjusted analysis, which is in line with previous meta-analyses [43–45]. Furthermore, evidence obtained from observational studies is an of great importance source for clinical practice where randomized clinical trials are unavailable or infeasible [46]. Unlike clinical trials through randomization to ensure patient characteristics are comparable across treatment and control groups,

observational studies usually attempt to adjust for confounding by multivariable regression and propensity score matching. We took the advantage of these approaches and applied adjusted ORs or HRs into our meta-analysis. Similarly, we found that outpatient metformin therapy was associated with decreased in-hospital mortality in the meta-analysis of studies with adjusted OR or adjusted HR. However, when only the propensity score-matched studies [22,23,28,30,33,36] were considered for outpatient metformin use and in-hospital mortality in patient with diabetes hospitalized for COVD-19, there was no significant association with reduced in-hospital mortality. A similar result was reached for inpatient metformin use and in-hospital mortality between the unadjusted and adjusted analyses. These discordant findings could be due to confounding bias, between-study heterogeneity and a small number of studies in the subgroup analysis. Therefore, further randomized clinical trials are needed to investigate the clinical benefits of metformin therapy in patient with diabetes hospitalized for COVD-19.

Several limitations are present in our study. First, owing to lack of randomized clinical trials, all studies included for analysis were retrospective studies. Although efforts were made to balance and control for potential confounding factors by multiple variable adjustments and propensity score matching, due to the inherent nature of retrospective observational studies, residual confounders are likely to exist and could not be balanced. Therefore, even meta-analysis of studies with adjusted ORs or HRs could be misleading. Second, the size of the retrospective studies that were included in the meta-analysis varied considerably, resulting in moderate-to-high between-study heterogeneity, as revealed by high $I^2$. Third, there were multiple sources of bias, resulting in high risk of bias in the meta-analysis. Confounding bias could be resulted from different patient characteristics, such as comorbidities, age, gender, and countries. Baseline of HbA1c, metformin dosage, duration of diabetes and treatment were often not reported in studies, contributing to heterogeneity and bias. Furthermore, different reported outcomes in studies could lead to selection and measurement bias. Fourth, there were very few studies investigating inpatient metformin use and in-hospital mortality. More studies are needed to perform a robust meta-analysis for inpatient metformin therapy and in-hospital mortality in patient with diabetes hospitalized for COVD-19.

In conclusion, our findings demonstrate that there is a significant association between the reduction of in-hospital mortality and outpatient metformin therapy for type 2 diabetes mellitus in patients hospitalized for COVID-19 and that inpatient metformin use is associated with a significant decrease of in-hospital mortality in the unadjusted analysis, but this mortality association does not retain after adjustments for confounding bias. Therefore, further randomized clinical trials are needed to provide clinical evidence regarding metformin therapy and in-hospital mortality in patient with diabetes hospitalized for COVD-19.

## Supporting information

**S1 Checklist. PRISMA 2020 checklist.**
(DOCX)

**S1 Fig. Sensitivity analysis of the studies included in meta-analysis in Fig 2A and 2C.**
(TIF)

## Author Contributions

**Conceptualization:** Zhiyuan Ma, Mahesh Krishnamurthy.

**Data curation:** Zhiyuan Ma, Mahesh Krishnamurthy.

**Formal analysis:** Zhiyuan Ma, Mahesh Krishnamurthy.

**Investigation:** Zhiyuan Ma, Mahesh Krishnamurthy.

**Methodology:** Zhiyuan Ma.

**Project administration:** Mahesh Krishnamurthy.

**Resources:** Mahesh Krishnamurthy.

**Supervision:** Mahesh Krishnamurthy.

**Validation:** Zhiyuan Ma.

**Writing – original draft:** Zhiyuan Ma.

**Writing – review & editing:** Zhiyuan Ma, Mahesh Krishnamurthy.

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
