## [Decision Letter · Decision Letter 0]

11 Nov 2022

PONE-D-21-37827Does Metformin Decrease Mortality in Patients with Type 2 Diabetes Mellitus Hospitalized for COVID-19? A Multivariable and Propensity Score-adjusted Meta-analysisPLOS ONE

Dear Dr. Krishnamurthy,

Thank you for submitting your manuscript to PLOS ONE. After careful consideration, we feel that it has merit but does not fully meet PLOS ONE’s publication criteria as it currently stands. Therefore, we invite you to submit a revised version of the manuscript that addresses the points raised during the review process. Please take particular consideration to overinterpretation of findings and use of language that suggests causation.

We look forward to receiving your revised manuscript.

Kind regards,

Jennifer A. Hirst, DPhil

Academic Editor

PLOS ONE

Journal Requirements:

Additional Editor Comments (if provided):

General comments – please add line numbers to help reviewers

The main finding of this paper is that although unadjusted analysis suggests a significant benefit of metformin, there is no effect of metformin on mortality after adjustment for confounders. This should be highlighted in the discussion, rather than the significant finding of the unadjusted analysis. Amend this in the abstract and main body of paper.

Methods.

The Prospero page needs updating as it indicates that the authors are still at the search stage.

Searches: These only identified a very small number of studies. Was a medical librarian consulted to assist with searches? Please provide full details of the search strategy, as the described searches would have missed articles. Consider including terms such as covid*, death, biguanide*, and metformin brand names.

Were terms for diabetes included in the search?

In the meta-analysis of adjusted data, please give details of methods used to allow odds ratio and hazard ratios to be combined. Were they assumed to be the same? If this is the case, then I would expect sensitivity analyses to be carried out to assess the impact of these approximations.

Results:

Outpatient metformin use and in-hospital mortality Page 6 – the following sentence needs some clarification: “There were 15 studies with or without attempt to adjust for confounding examining the association between outpatient metformin use and in-hospital mortality”

What is meant by with or without adjusting? Is this a repeat of the numbers in the previous paragraph?

The results section contains methods and interpretation – please ensure that methods are in the methods section and interpretation is moved to the discussion leaving results only to be reported in the results

Include details of quality assessment scores for each of the included studies and risk of bias.

Discussion

This needs to be structured, starting with the key findings.

Discuss risk of bias.

Reviewers' comments:

Reviewer's Responses to Questions

**Comments to the Author**

1. Is the manuscript technically sound, and do the data support the conclusions?

Reviewer #1: No

Reviewer #2: Yes

2. Has the statistical analysis been performed appropriately and rigorously? 

Reviewer #1: No

Reviewer #2: Yes

3. Have the authors made all data underlying the findings in their manuscript fully available?

Reviewer #1: No

Reviewer #2: No

4. Is the manuscript presented in an intelligible fashion and written in standard English?

Reviewer #1: Yes

Reviewer #2: Yes

5. Review Comments to the Author

Reviewer #1: What would be of particular interest is to be the first to draw some lessons from rather robust studies. This consists in fact of – to my knowledge – seventh meta-analysis.

Moreover, there are major drawbacks.

There are several errors:

- We are speaking about observational studies, not about interventional studies. Therefore the words “decrease” (mortality), as early as in the title, and “reduction”, appearing numerous times in the text, are not appropriate.

- The terms “outpatients” and “inpatients” are also inappropriate. The question should rather be phrased as whether or not the putative beneficial association between metformin treatment and COVID outcomes is due to metformin treatment prior to hospital admission, its continuation during the hospital stay, or both. Therefore the wording would be: “metformin discontinued”, “metformin continued”, “maintenance of metformin”, etc. (during a hospital stay).

- Title: what means “A Multivariable and Propensity Score-adjusted Meta-analysis?”

- The wording is often weak (one example, in the abstract: “Random effect models were applied for meta-analysis due to variation among studies”. In fact, the issue is not that of “variations” (between studies). Rather it is as following: the inclusion of a large number of covariates does not compensate for a lack of key variables (such as BMI, the estimated glomerular filtration rate, duration of diabetes, etc.). Given that, the studies with a low sample size, with a low number of covariates, and with a lack of the major covariates should not even be considered. It is already impossible to compare studies with either 172 or more than 2 million people with diabetes!

- There are just a few words about the putative protective effects for metformin; this is clearly not enough (what about effects on microvascular blood flow and pericytes, microvascular permeability, hemostasis, glycocalyx, hydrogen sulfide, ER stress, etc.?).

- The first lines of the Discussion do not deal with the main results.

- To put it simply, the compilation of short and/or weak studies cannot lead to a firm conclusion. And, as it happens, the by far largest study (more than 2 million people: Ojeda-Fernandez, Diabetes Obesity Metab 2022) is lacking.

- Given all these difficulties, the limitations subsection is far too weak.

Reviewer #2: Thank you for pursuing this question of metformin's role.

- You state that you assessed bias and refer to internal to analysis confounding issues as well as publication bias, but you do not characterize the latter. What were your systematic bias assessment results across the studies?

- Heterogeneity is quite high across pooled analyses. This should be discussed. For instance, (1) how was A1C and complicated diabetes accounted across studies; and, (2) in which studies was metformin already part of the patients' regimen versus added while in house or once diagnosis with SARS-CoV-2 was accomplished.

- The discussion and conclusion language should acknowledge more directly (depending upon your response to the previous questions) that metformin may indicate lower risk because being managed for diabetes... currently, your presented data and analyses does not support a direct effect and yet you close the article expressing only a caveat of pooled analyses statistical significance.

6. PLOS authors have the option to publish the peer review history of their article (what does this mean?). If published, this will include your full peer review and any attached files.

Reviewer #1: **Yes: **Jean-Daniel Lalau

Reviewer #2: No

---

## [Author Response · Author response to Decision Letter 0]

27 Nov 2022

1. “Please ensure that your manuscript meets PLOS ONE's style requirements, including those for file naming.” “please add line numbers to help reviewers”

Response: We have revised our manuscript according to PLOS ONE’s style requirements and added the line numbers.

2. “The main finding of this paper is that although unadjusted analysis suggests a significant benefit of metformin, there is no effect of metformin on mortality after adjustment for confounders. This should be highlighted in the discussion, rather than the significant finding of the unadjusted analysis. Amend this in the abstract and main body of paper.”

Response: We updated search and have performed new analysis. The results have been included in the section of Results and discussion. 

3. “The Prospero page needs updating as it indicates that the authors are still at the search stage.”

Response: We have updated the status in the PROSPERO. 

4. “Searches: These only identified a very small number of studies. Was a medical librarian consulted to assist with searches? Please provide full details of the search strategy, as the described searches would have missed articles. Consider including terms such as covid*, death, biguanide*, and metformin brand names.

Were terms for diabetes included in the search?”

Response: We did not consult librarian for assisting with the search. We did not include the keyword diabetes in the search. We manually screened diabetic patients in the studies after identification. Since our search was more than 1 year ago, we performed updated search using the expression “((SARS-CoV-2) OR (Covid-19)) AND ((Metformin) OR (Biguanides) OR (Fortamet) OR (Glumetza) OR (Glucophage) OR (Biomet)) AND ((Mortality) OR (Death))” as suggested. Three new studies have been included for analyses.

5. “In the meta-analysis of adjusted data, please give details of methods used to allow odds ratio and hazard ratios to be combined. Were they assumed to be the same? If this is the case, then I would expect sensitivity analyses to be carried out to assess the impact of these approximations.”

Response: Raw event data extracted from individual study were pooled using the Mantel-Haenszel approach and effect size (OR, HR) was pooled using the inverse-variance method. This has been described in the methods. We also included sensitivity analyses in Figure 2 and 3.

6. “Outpatient metformin use and in-hospital mortality Page 6 – the following sentence needs some clarification: “There were 15 studies with or without attempt to adjust for confounding examining the association between outpatient metformin use and in-hospital mortality”

What is meant by with or without adjusting? Is this a repeat of the numbers in the previous paragraph?”

Response: We have revised the text.

7. “The results section contains methods and interpretation – please ensure that methods are in the methods section and interpretation is moved to the discussion leaving results only to be reported in the results”

Response: We have revised the results. 

8. “Include details of quality assessment scores for each of the included studies and risk of bias.”

Response: the Newcastle-Ottawa Scale scores have been included in Table 1. 

9. “This needs to be structured, starting with the key findings.

 Discuss risk of bias.”

Response: We have revised the section of discussion and discussed more on bias. 

Reviewer #1: 

1. “What would be of particular interest is to be the first to draw some lessons from rather robust studies. This consists in fact of – to my knowledge – seventh meta-analysis.”

Response: Thank you for your comments. Of note, we performed this study and submitted it more than a year ago. In this study, we aimed to address whether outpatient (prior to admission) or inpatient metformin (during hospitalization) therapy offers low in-hospital mortality in patients with type 2 diabetes mellitus hospitalized for COVID-19 especially by multivariable and propensity score-adjusted effect sizes (OR and HR). We discussed the issues in the previously published meta-analysis in the second paragraph of the section of the discussion. We think our study is interesting. 

2. “We are speaking about observational studies, not about interventional studies. Therefore the words “decrease” (mortality), as early as in the title, and “reduction”, appearing numerous times in the text, are not appropriate.”

Response: Thank you for your comments. We agree that we did a meta-analysis of observational studies. We have revised our title. But we believe metformin therapy is interventional and treatment, so “decrease” and “reduction is related to metformin therapy not to study types. 

3. “The terms “outpatients” and “inpatients” are also inappropriate. The question should rather be phrased as whether or not the putative beneficial association between metformin treatment and COVID outcomes is due to metformin treatment prior to hospital admission, its continuation during the hospital stay, or both. Therefore the wording would be: “metformin discontinued”, “metformin continued”, “maintenance of metformin”, etc. (during a hospital stay).”

Response: Thank you for your suggestion. We have specifically clarified “outpatient” and “inpatient” metformin use in the introduction. 

4. “what means “A Multivariable and Propensity Score-adjusted Meta-analysis?””

Response: Meta-analysis of studies with adjusted effect size derived from multivariable regression or propensity score matching. 

5. “The wording is often weak (one example, in the abstract: “Random effect models were applied for meta-analysis due to variation among studies”. In fact, the issue is not that of “variations” (between studies). Rather it is as following: the inclusion of a large number of covariates does not compensate for a lack of key variables (such as BMI, the estimated glomerular filtration rate, duration of diabetes, etc.). Given that, the studies with a low sample size, with a low number of covariates, and with a lack of the major covariates should not even be considered. It is already impossible to compare studies with either 172 or more than 2 million people with diabetes!”

Response: Thank you for your comments. We have revised the text. The random-effects model is conventionally used to address between-study heterogeneity. We did unadjusted and adjusted analyses to address other bias. we excluded studies with the number of subjects less than 50. 

6. “There are just a few words about the putative protective effects for metformin; this is clearly not enough (what about effects on microvascular blood flow and pericytes, microvascular permeability, hemostasis, glycocalyx, hydrogen sulfide, ER stress, etc.?).”

Response: We have revised the text and added the protected effects on microvessels. 

7. “The first lines of the Discussion do not deal with the main results.”

Response: We have revised the section of discussion.

8. “To put it simply, the compilation of short and/or weak studies cannot lead to a firm conclusion. And, as it happens, the by far largest study (more than 2 million people: Ojeda-Fernandez, Diabetes Obesity Metab 2022) is lacking.”

Response: We completed our search before Oct 1st, 2021 and submitted our paper on Nov. 29th, 2021. Now We completed new search and have included the above study along with 2 more studies.

9. “Given all these difficulties, the limitations subsection is far too weak.”

Response: We have revised the limitation subsection. 

Reviewer #2:

1. “You state that you assessed bias and refer to internal to analysis confounding issues as well as publication bias, but you do not characterize the latter. What were your systematic bias assessment results across the studies?”

Response: We have performed the analyses of publication bias using funnel plots and Egger’s tests. We also included the Newcastle-Ottawa Scale scores in Table 1.

2. “Heterogeneity is quite high across pooled analyses. This should be discussed. For instance, (1) how was A1C and complicated diabetes accounted across studies; and, (2) in which studies was metformin already part of the patients' regimen versus added while in house or once diagnosis with SARS-CoV-2 was accomplished.”

Response: We have revised the limitation subsection. 

3. “The discussion and conclusion language should acknowledge more directly (depending upon your response to the previous questions) that metformin may indicate lower risk because being managed for diabetes... currently, your presented data and analyses does not support a direct effect and yet you close the article expressing only a caveat of pooled analyses statistical significance.”

Response: We have performed new analyses with more studies. We have revised the results and conclusions.

---

## [Decision Letter · Decision Letter 1]

30 Jan 2023

PONE-D-21-37827R1Is Metformin Use Associated with Low Mortality in Patients with Type 2 Diabetes Mellitus Hospitalized for COVID-19? A Multivariable and Propensity Score-adjusted Meta-analysisPLOS ONE

Dear Dr. Krishnamurthy,

Thank you for submitting your manuscript to PLOS ONE. After careful consideration, we feel that it has merit but does not fully meet PLOS ONE’s publication criteria as it currently stands. Therefore, we invite you to submit a revised version of the manuscript that addresses the points raised during the review process.

We look forward to receiving your revised manuscript.

Kind regards,

Jennifer A. Hirst, DPhil

Academic Editor

PLOS ONE

Journal Requirements:

Reviewers' comments:

Reviewer's Responses to Questions

**Comments to the Author**

1. If the authors have adequately addressed your comments raised in a previous round of review and you feel that this manuscript is now acceptable for publication, you may indicate that here to bypass the “Comments to the Author” section, enter your conflict of interest statement in the “Confidential to Editor” section, and submit your "Accept" recommendation.

Reviewer #1: All comments have been addressed

Reviewer #2: (No Response)

2. Is the manuscript technically sound, and do the data support the conclusions?

Reviewer #1: Yes

Reviewer #2: Partly

3. Has the statistical analysis been performed appropriately and rigorously? 

Reviewer #1: Yes

Reviewer #2: Yes

4. Have the authors made all data underlying the findings in their manuscript fully available?

Reviewer #1: Yes

Reviewer #2: Yes

5. Is the manuscript presented in an intelligible fashion and written in standard English?

Reviewer #1: Yes

Reviewer #2: Yes

6. Review Comments to the Author

Reviewer #1: I must admit that I am not very interested in meta-analyses but I would also highlight the great work done by the authors for improving their manuscript.

I agree with all modifications but one: as long as studies are observational, it is never possible to speak about “reduction” (of mortality – because of metformin) (lines 21, 158, 211, 266, 269…), and even less about a “trend toward a reduction” (L. 158: non-scientific).

Reviewer #2: Thank you for your work addressing editorial and reviewer comments. The manuscript is improved.

- Consider doing a quick sensitivity analysis where you remove all studies that have a 6 or lower Newcastle-Ottowa score (indicating high risk). Those studies are substantially smaller than the studies that you scored more highly and so a single study removal method for sensitivity analysis would likely be insufficient to discern the effect by class.

- The bigger issue for me is that the manuscript is written in a way that suggests that metformin was applied for the purpose of COVID-19 therapy. That seems unlikely, particularly in the one category where you observed a multi-variate statistically significant effect. This matters in how readers (even technically grounded ways) understand the role of metformin here. You also seem to avoid talking about DM management in your background as a reason for exploring the effect. This needs to be addressed through-out. The Limitations section additions relevant to my earlier comments are appropriate. What I am talking about now is how the question is framed and the discussion and conclusions are proffered. I suspect that you simply have taken for granted that readers will understand this difference, but you do not state it, and your language suggests direct therapeutic application for the purpose of COVID-19 in a milieu where it has been among several problematic narratives avoiding more rounded approaches to patient management. This can be stated succinctly in the intro, discussion, and conclusion sections and easily corrected.

7. PLOS authors have the option to publish the peer review history of their article (what does this mean?). If published, this will include your full peer review and any attached files.

Reviewer #1: **Yes: **Jean-Daniel Lalau

Reviewer #2: No

---

## [Author Response · Author response to Decision Letter 1]

4 Feb 2023

Reviewer #1: 

1. “I must admit that I am not very interested in meta-analyses but I would also highlight the great work done by the authors for improving their manuscript.

I agree with all modifications but one: as long as studies are observational, it is never possible to speak about “reduction” (of mortality – because of metformin) (lines 21, 158, 211, 266, 269…), and even less about a “trend toward a reduction” (L. 158: non-scientific)”

Response: Thank you for your comment. As you pointed out that this is a meta-analysis of observational studies, we have revised the text and made it clear throughout the study that there is an association between metformin use and hospital mortality, but not direct causation. The word “reduction” is used as a noun or adjective not as a verb which may imply direct causation.

Reviewer #2: 

1. “Consider doing a quick sensitivity analysis where you remove all studies that have a 6 or lower Newcastle-Ottowa score (indicating high risk). Those studies are substantially smaller than the studies that you scored more highly and so a single study removal method for sensitivity analysis would likely be insufficient to discern the effect by class”

Response: We have added additional analysis regarding this (Supp fig. 1A). 

2. “The bigger issue for me is that the manuscript is written in a way that suggests that metformin was applied for the purpose of COVID-19 therapy. That seems unlikely, particularly in the one category where you observed a multi-variate statistically significant effect. This matters in how readers (even technically grounded ways) understand the role of metformin here. You also seem to avoid talking about DM management in your background as a reason for exploring the effect. This needs to be addressed through-out. The Limitations section additions relevant to my earlier comments are appropriate. What I am talking about now is how the question is framed and the discussion and conclusions are proffered. I suspect that you simply have taken for granted that readers will understand this difference, but you do not state it, and your language suggests direct therapeutic application for the purpose of COVID-19 in a milieu where it has been among several problematic narratives avoiding more rounded approaches to patient management. This can be stated succinctly in the intro, discussion, and conclusion sections and easily corrected.”

Response: Thank you for your comment. We have revised the text.

---

## [Editor Report · Decision Letter 2]

10 Feb 2023

Is Metformin Use Associated with Low Mortality in Patients with Type 2 Diabetes Mellitus Hospitalized for COVID-19? A Multivariable and Propensity Score-adjusted Meta-analysis

PONE-D-21-37827R2

Dear Dr. Krishnamurthy,

We’re pleased to inform you that your manuscript has been judged scientifically suitable for publication and will be formally accepted for publication once it meets all outstanding technical requirements.

Kind regards,

Jennifer A. Hirst, DPhil

Academic Editor

PLOS ONE
---

## [Editor Report · Acceptance letter]

14 Feb 2023

PONE-D-21-37827R2 

Is Metformin Use Associated with Low Mortality in Patients with Type 2 Diabetes Mellitus Hospitalized for COVID-19? A Multivariable and Propensity Score-adjusted Meta-analysis 

Dear Dr. Krishnamurthy:

I'm pleased to inform you that your manuscript has been deemed suitable for publication in PLOS ONE. Congratulations! Your manuscript is now with our production department. 

Kind regards, 

on behalf of

Dr. Jennifer A. Hirst 

Academic Editor

PLOS ONE